# Fungal Osteomyelitis: A Systematic Review of Reported Cases

**DOI:** 10.3390/microorganisms11071828

**Published:** 2023-07-17

**Authors:** Erika Asperges, Giuseppe Albi, Francesco Truffelli, Andrea Salvaderi, Flavia Puci, Aurelia Sangani, Valentina Zuccaro, Valeria Scotti, Paolo Orsolini, Enrico Brunetti, Raffaele Bruno

**Affiliations:** 1S.C. Malattie Infettive I Fondazione IRCCS Policlinico San Matteo, 27100 Pavia, Italy; francesco.truffelli01@universitadipavia.it (F.T.); andrea.salvaderi01@universitadipavia.it (A.S.); flavia.puci01@universitadipavia.it (F.P.); aurelia.sangani01@universitadipavia.it (A.S.);; 2Department of Electrical, Computer and Biomedical Engineering, Università di Pavia, 27100 Pavia, Italy; 3Dipartimento di Scienze Clinico-Chirurgiche, Diagnostiche e Pediatriche—Università di Pavia, 27100 Pavia, Italy; 4UOSD Grant Office, TTO e Documentazione Scientifica, Fondazione IRCCS Policlinico San Matteo, 27100 Pavia, Italy; 5Dipartimento di Medicina Interna e Terapia Medica, Università di Pavia, 27100 Pavia, Italy

**Keywords:** fungal osteomyelitis, fungal spondylodiscitis, *Candida*, *Aspergillus*, *Crytpococcus*, *Mucormycosis*, *Scedoscporium*, *Coccidioides*

## Abstract

Fungal osteomyelitis is considered a rare disease, and the published literature mainly comprises case reports, case series and narrative reviews. A systematic review was undertaken to provide a practice-based global perspective on this disease, focusing on epidemiology and treatment strategies. We searched MEDLINE, EMBASE and Cochrane Library between the 3rd and 8th of March 2023 using a predefined search string. We included studies with at least one patient with a diagnosis of fungal osteomyelitis published before the 1st of January 2023. We included all study designs except for reviews, and we excluded non-English languages and grey literature. After exclusion, 678 studies, mostly case reports, were included. Descriptive analysis was performed on 1072 patients. The most common aetiological agent was *Aspergillus* (26.5%), followed by *Candida* (20.7%) and *Mucor* (16.8%), and the bones most frequently involved were the vertebrae. We described the characteristics of patients divided by site of infection, and we found that diabetes mellitus, disseminated fungal infection, surgery and local lesion were major risk factors. We also successfully associated duration of treatment with outcome. We provided a general overview of this rare disease, and we highlighted the need for high-quality investigations on the subject.

## 1. Introduction

Fungi are pervasive microorganisms, but only about 200 species are pathogenic to humans, with a dozen causing 90% of all human mycoses [1]. Fungi often cause indolent infections in immunocompetent individuals, but they have a propensity to quickly disseminate and can rapidly become fatal in immunosuppressed hosts. With a rising number of such patients, the incidence of invasive fungal infections is also increasing, meaning that timely diagnosis and treatment will become even more important [2]. 

Despite this, fungal osteomyelitis is considered a rare disease and is often overlooked when compiling differential diagnosis. In fact, symptoms are often subacute and mimic those of other aetiologies, which can lead to substantial delays in treatment [3]. This situation is reflected by the literature, which mainly comprises case reports, case series and a few narrative reviews. 

The current systematic review was undertaken to provide a practice-based global perspective on this disease, exploring epidemiology characteristics (age, gender, geographical origin, main risk factors, aetiology), as well as management options such as diagnostic and treatment strategies. We used the CoCoPop framework for studies on prevalence and aetiologies (Condition: fungal osteomyelitis; Context: worldwide; Population: no limitations) [4]. Using individual-level data of case reports, we explored potential associations between outcomes of interest (i.e., death, recovery and recurrence) and surgical treatment and treatment duration.

## 2. Materials and Methods

This study follows the Preferred Reporting Items for Systematic Reviews and Meta-Analyses (PRISMA) guidelines [5] and was registered in the PROSPERO online database (number CRD42023401013).

### 2.1. Data Sources and Management

We searched MEDLINE, EMBASE, and Cochrane Library using a predefined search string on the 3 March 2023 (MEDLINE) and on the 8 March 2023 (EMBASE and Cochrane Library). Search queries are available in Appendix A. 

All abstracts and full texts were imported into and managed in the web database Rayyan [6] and shared between senior researchers (EA, VZ), junior researchers (FT, AnS, FP, AuS) and the librarian (VS). Abstract and full-text review was performed independently by the junior researchers. Disagreements were resolved by the senior researchers.

### 2.2. Inclusion and Exclusion Criteria

We included all studies with at least one patient with a diagnosis of fungal osteomyelitis published before the 1 January 2023. All study designs were included except for reviews. We excluded studies in languages different than English and grey literature like unpublished articles, conference abstracts and any non-peer-reviewed works. Given the scarcity of literature on the subject, we did not specify additional exclusion criteria in order to collect the largest possible number of patients. 

### 2.3. Data Extraction

Individual-level data on patients were extracted by EA, VZ, FT, AnS, FP and AuS onto a dedicated spreadsheet. Extracted data include first author, publication year, language, study design, timeline (prospective or retrospective), total number of patients included in the study and the number of patients with fungal osteomyelitis. For each patient with fungal osteomyelitis, we collected age (mean age if more than one patient), sex, country of origin, aetiology (e.g., *Candida*, *Scedosporium*…), risk factors (e.g., diabetes, trauma, haematological disease), site of infection (e.g., sternum, limbs…), diagnosis (culture, histology, empiric), other mycological diagnostic methods (cryptococcal, coccidioidal and *Histoplasma* antigen tests, *Candida*-specific serologies, beta-D-glucan, galattomannan antigen), radiological evidence of disease, time to diagnosis (less than 1 month, between 1 and 6 months, more than 6 months), treatment (antifungal drugs and duration, surgery) and outcome (death, resolution, recurrence/chronicization). 

### 2.4. Risk of Bias

Given the various study designs included in this review, bias was assessed qualitatively using Joanna Briggs Institute’s critical appraisal checklists. Checklists for case reports [7], case series [8], cohort studies [7] and randomised controlled trials [9] were used as appropriate. Quality was considered adequate if all the items in the checklist were answered “yes”, but studies with inadequate quality were not discarded in order to collect all available data on this rare condition. Quality assessment was performed independently by EA, VZ, FT, AnS, FP and AuS with supervision by EA. 

### 2.5. Data Synthesis

No formal data synthesis was applicable to the present review. Given the nature of the included studies and data, individual-level qualitative variables were described using raw values and frequencies, while quantitative variables were described using measures of central tendency as appropriate. Studies were excluded from the descriptive analysis if specific patient data (patients’ numbers, age, gender and site of infection) were missing. Associations between the outcomes of interest, surgical treatment and treatment duration were assessed with logistic regression using individual-level data in order to avoid aggregation bias. In addition, the difference in terms of site of infections and aetiology were assessed with two-tailed Chi-squared test or Fisher exact test, for both the total population and immunocompromised population. *p*-value less than 0.05 was considered significative. Studies from which individual-level data could not be extrapolated (e.g., case series) were excluded from these analyses. 

## 3. Results

After removal of 315 duplicates among the three databases, the titles and abstracts of 1317 records were manually screened for full-text assessment. Overall, 940 records were selected for full-text evaluation, but the manuscript could not be retrieved for 206 records; thus, eligibility was assessed for 734. Reasons for exclusion included wrong topic (37), wrong article type (12 review and 3 grey literature), non-English language (2) and being duplicates (2). Reasons for exclusion and citations are available in Appendix A. A total of 678 studies were included for data extraction. Citations and studies’ characteristics are provided in Appendix A. From these, 88 studies with missing specific patient data were excluded from descriptive and association analyses. A further 89 studies that included multiple patients were excluded only from association analyses in order to avoid aggregation bias. 

The PRISMA flowchart that summarises all the steps performed for the identification and for the screening of the included studies is reported in Figure 1.

### 3.1. Studies’ Characteristics

The studies included in this review were published between 1953 and 2022 (Figure 2A) and come from 59 different countries, with the largest contribution coming from the USA, with 226 studies, followed by India with 62 and UK with 28 studies (Figure 2B).

The study types included mainly case reports (79.2%) and case series (17.4%). Indeed, the median number of patients per study is 1, with a range from 1 to 50. The study design was almost exclusively retrospective, since only 9 out of the 678 studies were prospective (1.3%). Risk of bias for each study is presented in the Appendix A. We judged 83.3% of studies to be of adequate quality. The main reasons for exclusion were missing information on clinical progress and unclear inclusion criteria. 

### 3.2. Patients’ Characteristics

We collected data from 1072 patients and reported their characteristics in Table 1. There were 735 males (68.6%) and 337 females (31.4%). The median age was 45 years, with a range from 0.2 to 92 years. The majority of studies and patients focused on *Aspergillus* infection, with *Candida* in second place. In third place, there was cryptococcal infection, but overall, there were more patients with mucormycosis, as described in Figure 3A,C. The three more frequent risk factors were surgery/local lesion, diabetes mellitus and disseminated fungal infection. Diagnosis from bone culture or histology and radiology was obtained in >70% in cases, but only 26% of patients reached it within 1 month. Other microbiological tests included species-specific tests (supporting diagnosis of *Cryptococcus* in 44.1% of cases, of *Coccidioides* in 51%, of *Histoplasma* in 28.0% of cases, of *Candida* in 1.3%) and non-specific tests (beta-D-glucan and galattomannan antigen), supporting diagnosis of *Candida* and *Aspergillus* in much lower percentages (3.4% and 7.2%, respectively). Polymerase chain reaction was only very rarely used for diagnosis. In total, 72.8% of patients achieved full recovery. Surgery was performed in 65.8% of patients, and median treatment was 4 months (range: 0–48), with 23 patients needing chronic therapy. Treatment strategies are summarised in Figure 4, showing a net preference for Amphotericin B and azoles.

### 3.3. Sites of Infection

Study population was divided according to site of infection. The majority of articles described limb osteomyelitis, but the majority of patients had vertebral osteomyelitis, followed by the extremities and the cranium. We present a comparative description of the sites of infection below and in Table 2.

Basicranium: The prevalent aetiologies were *Aspergillus* (43.9%) and *Mucormycosis* (31.0%), followed by a good margin by *Candida* species (10.3%). Several other species were detected, though in lower numbers. Nearly half of the patients were diabetic, and more than half had an immunosuppressive condition, either due to an oncohaemotological disease (33.6%) or of other origin (26.7%). Other relevant risk factors, with a prevalence over 20%, included disseminated fungal infection and surgery or lesion of the basicranium. Diagnosis was relatively quick, with only 19.8% of patients going over 1 month. The disease was fatal in 20.7% of patients, making this site of infection the most severe. Surgery was necessary in 76.7% of cases, the highest among the different sites, but the range of treatment duration was only 0–18 months (median = 3.7), probably reflecting the high mortality.Splanchnocranium: Prevalent aetiologies (*Aspergillus*, *Mucormycosis*, *Candida*) were the same as the basicranium, but, despite a much bigger number of cases, other species were much rarer. Diabetes had nearly the same prevalence, but while immunosuppression in general also played a role, oncohaematological diseases were less prevalent by half (17.1%). COVID-19, moreover, was a recent significant risk factor, being present in 16.5% of patients, while it was rare in other cranial bone infections and absent as a risk factor in other bones. Diagnosis, either by culture or histology, was reached much less frequently than in basicranium infection, and the delay was also much more marked, with 32% of patients reaching a diagnosis after over 1 month. Mortality, however, was halved, with only 10.2% of cases resulting in death.Other cranial bones: More than half were caused by *Aspergillus* species, with *Mucormycosis* playing a smaller role and *Candida* causing only 5.6% of cases. *Blastomyces* and *Cryptococcus*, on the other hand, caused many cases compared to the other cranial bones (7.3% and 6.2%, respectively). The relative prevalence of risk factors was also significantly different, with disseminated fungal infection taking first place. Despite the low prevalence, HIV was more prevalent in this site of infection than in any other. A high percentage of patients (8.4%) did not reach a diagnosis before 6 months, and mortality was nearly as high as in basicranium infections.Extremities (except foot): This site of infection presented the biggest variability in aetiologies. *Aspergillus* caused nearly one-fourth of infections (24.3%), followed by *Coccidioides* (11.5%), but *Blastomyces*, *Candida*, *Cryptococcus*, *Mucormycosis* and *Paracoccidioides* all caused at least 5% of infections, and *Fusarium*, *Histoplasma* and *Scedosporium* all caused at least 2%. The “Other” category was more prevalent here than in any other site (6.3%). The main risk factors appeared to be surgery and local lesions (27.7%) and disseminated infection (26.7%). Immunosuppression, diabetes, oncohaematological diseases, transplantation and prolonged hospitalisation were also present in at least 10% of cases. Diagnosis through other microbiological methods was relatively common (17.5%) due to the abundance of *Coccidioides*. Death was relatively rare (12.0%).Foot: This site also presented a large variability in aetiologies, with *Aspergilus*, *Coccidioides* and *Paracoccidioides* taking centre stage. *Fusarium* caused 8.7% of infections. Risk factors did not differ from those of the other extremities, even though, surprisingly, local lesions and surgery decreased to 11.6%. Also surprising were the low percentage of patients who reached a diagnosis in less than a month (10.9%, lower than that of all other sites) and the relatively low number of deaths (9.4%).Pelvis and hip: *Aspergillus*, *Coccidioides*, *Candida* and *Paracoccidioides* caused the majority of infections. The main risk factors were disseminated fungal infection and surgery or local lesion, while diabetes and bed sores did not appear to be relevant risk factors, with percentages of 4.6% and 0.6%, respectively. Indirect microbiological diagnosis was reached in 18.9% of patients. This site showed the highest risk for recurrence and chronicization (18.3%). Surgery was performed more rarely than in other sites (45.1%).Ribs and sternum: This site showed the highest disproportion in the male:female ratio (75.8 and 24.2% respectively). *Aspergillus* caused nearly half of the cases, but *Candida* and *Coccidioides* were also common. *Paracoccidioides* and *Cryptococcus* both caused 7.8% of cases, and *Blastomyces* caused 5.7%. Disseminated fungal infection, local surgery or lesion and immunosuppression were the main risk factors.Shoulder: Following *Aspergillus*, the most common aetiology was *Paracoccidioides* (24.3%), while *Coccidioides* only comprised 5.2% of cases. Shoulder infection had the highest rate of recurrence or chronicization following the hip (17.4%), and it also had the lowest range of treatment duration (1–15 months) and rate of surgery (47.8%).Spine: About one-third of infections were caused by *Aspergillus* and another third by *Candida*. There was a relatively high percentage of intravenous drug users (7.6%) compared to other sites. This site had the highest percentage of direct microbiological diagnosis (83.9%) but also the highest percentage of cases with time to diagnosis exceeding a month (42.8%).

### 3.4. Aetiologies

A comparative analysis of aetiologies in presented in Appendix A. 

### 3.5. Factors Associated with Outcome

Results of logistic regressions showing association of outcome with variables of interest are shown in Table 3. 

Treatment duration (overall median 4 months) showed an association with both mortality and survival, with longer treatment being protective (OR 1.1, CI 1.1–1.3; *p* < 0.001) for survival and shorter treatment being predictive of mortality (OR 0.54, CI 0.38–1.3, *p* < 0.01). Surgical treatment showed a trend toward a positive outcome, being protective against mortality (OR 0.69) and improving the chances of recovery (OR 1.54) even though neither reached statistical significance. 

*Aspergillus* infection was strongly associated with mortality (OR 2.7, CI 1.5–4.8, *p* < 0.001). *Fusarium* infection also showed a tendency to association with mortality (OR 3.1, CI 0.7–11.2) but failed to reach statistical significance, probably because of the low number of cases. No specific aetiology was associated with survival. 

Regarding sites of infection, vertebral infection decreased the chance of survival (OR 0.5, CI 0.3–0.9, *p* < 0.01). 

Risk factors for chronicity or recurrence were histoplasmosis, shoulder infection and time to diagnosis > 1 month, especially if > 6 months (OR 3.0, CI 1.6-5.5, *p* < 0.001). 

A condition of immune suppression was not associated with survival or mortality. 

A second logistic regression was performed excluding the 54 articles of insufficient quality. The results are shown in Appendix A. In this analysis, the association of *Fusarium* infection with death gained significance (OR 4.97, CI 0.99–20.89, *p* = 0.03), while the association of shoulder infection with chronicity or recurrence lost significance (*p* > 0.05). The remaining results did not change. 

## 4. Discussion

The aim of this systematic review was to provide a general overview of fungal osteomyelitis, which is a very rare disease. To this end, this review describes 1072 patients collected from 678 studies, mostly case reports. The majority of patients had *Aspergillus* infection, and the bones most frequently involved were the vertebrae. 

Initial analysis focused on patients’ characteristics and later on the sites of infection. Diabetes mellitus, disseminated fungal infection, surgery and local lesion were major risk factors. The risk posed by disseminated fungal infection might explain the high prevalence of vertebral osteomyelitis, as the spine is a preferred site of haematogenous spread [10]. Traditional conditions of immune suppression (oncohaematological diseases, transplant patients, HIV) also represented significant risk factors, but were overall less represented than expected, and the logistic regression did not demonstrate a significant impact of these risk factors on mortality. The comparative analysis of aetiologies, however, did show that HIV was more frequent in cryptococcal infections (4.3%) than in other classical HIV-associated fungal infections (e.g., 2.7% for *Candida*, 0% for *Pneumocystis jirovecii*, for which no cases were found). The high prevalence of local surgery and lesions also proved interesting, in the authors’ opinion, as it underlines the need for considering fungal infection even in immunocompetent patients. 

The rarity of empirical treatment and the high rate of diagnosis through histology or culture might indicate a reluctance to consider fungi as a possible cause of osteomyelitis without microbiological data. However, species-specific microbiological tests proved invaluable for cryptococcal and coccidioidal infection, while PCR and non-species-specific tests, commonly used for other fungi (e.g., galattomannan antigen tests), were rarely represented, even though this might be due to the age of many articles. 

These results are consistent with available literature, which is, however, mostly composed of narrative reviews and reviews of case reports [11,12]. Gamaletsou et al., for example, agree in their review of cases with these authors’ conclusion of haematogenous spread being the main cause of spinal infections, but their analysis was limited to one aetiology (*Aspergillus*) [13]. Koehler et al., who described a small number of cases of rare fungi, also underlined the important role of direct inoculation in immunocompetent patients [14], while Kohli et al. confirmed the small role played by HIV infection [15]. 

This work also shed a light on some open questions. For example, the data did not allow resolution of the scientific debate regarding the role of bedsores, as the prevalence of this risk factor was very low. 

Another open question is treatment duration and the role of surgery. The logistic regression successfully associated a longer treatment with survival and a shorter treatment with increased mortality. Although statistical significance was not reached, the surgical approach appeared to have a suggested protective trend against mortality. Further consideration of drugs of choice could not be proposed given the wide variety of aetiologies and drugs, even though Amphotericin B clearly emerged as the preferred choice. 

The regression analysis also underlined how diagnosis delay is extremely significant in this disease and how this is associated with recurrence and chronicity. 

This work suffers from several limitations, the most relevant of which is poor quality of existing literature on fungal osteomyelitis: this review found almost exclusively case reports and case series, and nearly 20% of these were of inadequate quality. In fact, the checklists used by the authors to assess bias were able to highlight deficiencies in all four domains of bias suggested by Murad et al. for case reports and case series [16]: selection, ascertainment (of the outcome), causality (i.e., follow-up) and reporting. The most relevant biases were selection and reporting, with causality playing a smaller role and ascertainment almost none at all, given the nearly ubiquitous certainty of the diagnosis. For example, in the selection domain, the authors believe that *Candida* infections, which are considered relatively common, are reported less frequently than the others, and in the causality and reporting domains, many articles failed to adequately describe their cases. 

Another limitation is that, in the effort to identify possible associations with the outcome, aggregated data were excluded from the analysis. Moreover, the impact of treatment duration on the outcome is probably influenced by the bias of early mortality for severe infections, without having reached the end of therapy. 

Despite these limitations, this review provides a general overview of this rare disease based on more than 1000 patients, and the limitations themselves highlight the need for high-quality investigations on the subject, however difficult the collection of data might be.

## Figures and Tables

**Figure 1 microorganisms-11-01828-f001:**
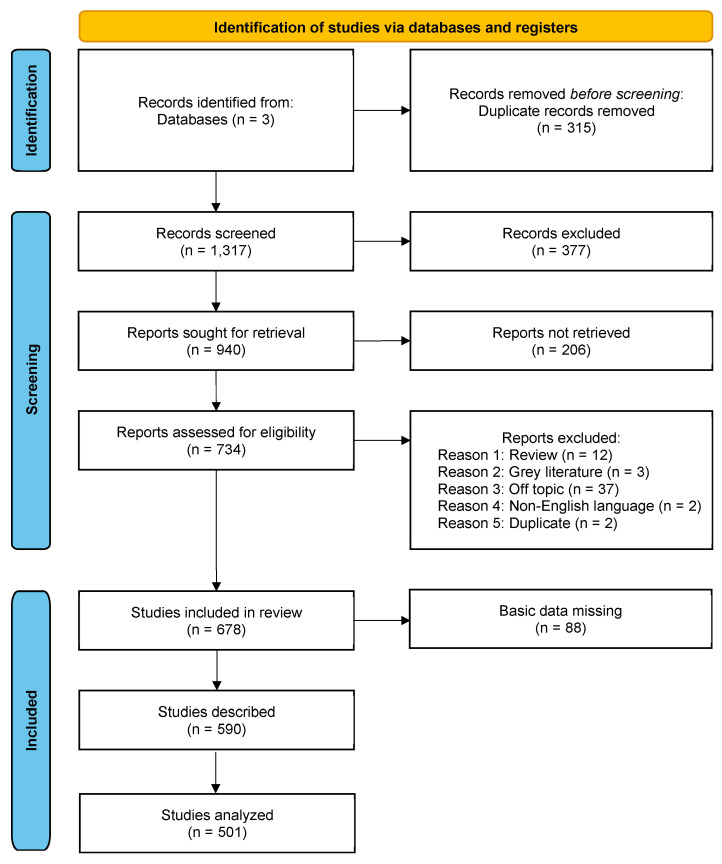
PRISMA flow chart for studies identification, screening and inclusion.

**Figure 2 microorganisms-11-01828-f002:**
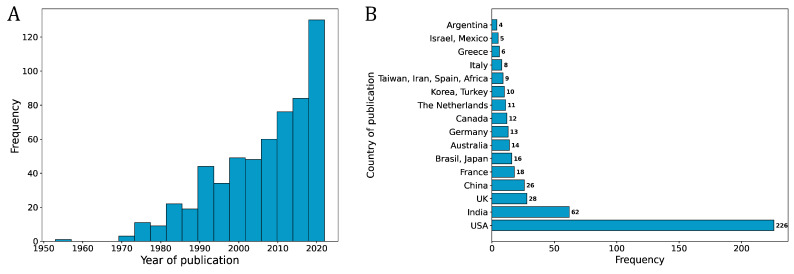
Histogram showing the year of publication (range = 1953–2022) frequency distribution for the 590 selected studies (**A**) and histogram showing the frequency distribution of the countries of origin of the patients, excluding countries contributing 3 or fewer patients (**B**).

**Figure 3 microorganisms-11-01828-f003:**
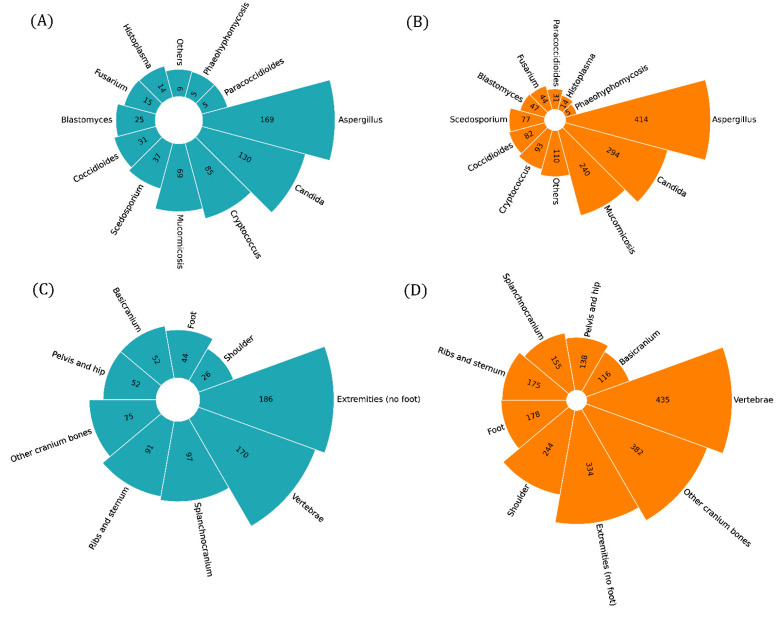
Circular bar plots showing the frequency distribution of the aetiological agents among the selected studies (**A**), the aetiological agents among the patients (**B**), the site of infection among the selected studies (**C**) and the site of infection among the patients (**D**).

**Figure 4 microorganisms-11-01828-f004:**
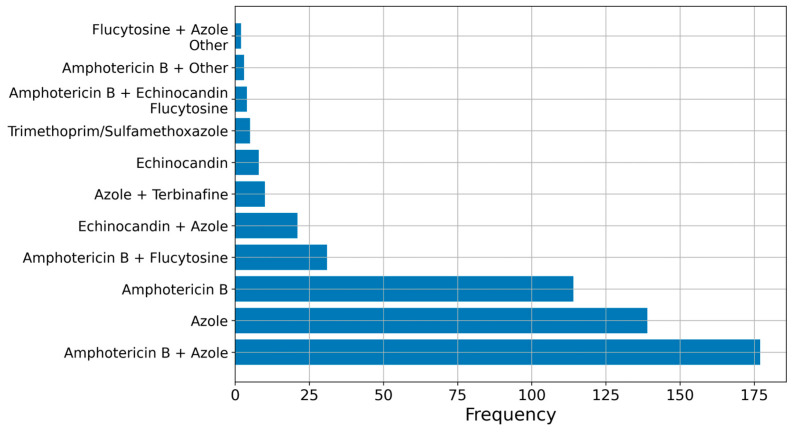
Bar plot showing the frequency distribution of antifungal treatment adopted in each study. Note that some studies included a combined treatment of two or more classes of antifungals.

**Table 1 microorganisms-11-01828-t001:** Patients’ characteristics. Numerical variables are presented as median and range, while categorical variables are presented as counts and percentage. CGD: chronic granulomatous disease; CVC: central venous catheter; IDU: intravenous drug users.

Patients (n = 1072)	
Age, years	45 [0.2–92]
Gender Male Female	735 (68.6) 337 (31.4)
Aetiological agent *Aspergillus* *Blastomyces* *Candida* *Coccidioides* *Cryptococcus* *Fusarium* *Histoplasma* *Mucormycosis* *Paracoccidioides* *Phaeohyphomycosis* *Scedosporium* Other	284 (26.5) 47 (4.4) 222 (20.7) 60 (5.6) 93 (8.7) 21 (1.9) 14 (1.3) 180 (16.8) 31 (2.9) 5 (0.5) 40 (3.7) 36 (3.3)
Site of infection Basicranium Extremities (except foot) Foot Other cranium bones Pelvis and hip Ribs and sternum Shoulder Splanchnocranium Vertebrae	66 (6.2) 290 (27) 48 (4.5) 91 (8.5) 62 (5.8) 118 (11) 35 (3.3) 265 (24.7) 318 (29.7)
Risk factors Bedsores CGD COVID-19 CVC Diabetes mellitus Disseminated fungal infection Fungemia HIV Hospitalised < 6 months IDU Oncohaematological disease Other immunodepression Parental nutrition Prosthesis Transplant Surgery/local lesion	5 (0.5) 15 (1.4) 57 (5.3) 72 (6.7) 261 (24.3) 234 (21.8) 74 (6.9) 23 (2.1) 155 (14.5) 43 (4) 149 (13.9) 198 (18.5) 22 (2) 31 (2.9) 91 (8.5) 288 (26.9)
Diagnosis Empirical Microbiological isolate from bone Histological Other microbiological methods Radiological	23 (2.1) 878 (81.9) 785 (73.2) 125 (11.7) 826 (77)
Time to diagnosis 0–1 month 1–6 months >6 months	279 (26) 381 (35.5) 97 (9)
Outcome Death Recovery Recurrence/chronicity	113 (10.5) 781 (72.8) 118 (11)
Treatment Duration, months Known antifungal ≥ 1 Known antifungal ≥ 2 Empirical Surgery	4 [0–48], 23 * 877 (81.8) 533 (49.7) 35 (3.3) 705 (65.8)

* Patients who required chronic treatment.

**Table 2 microorganisms-11-01828-t002:** Patients’ characteristics according to the site of infection. Numerical variables are presented as median and range, while categorical variables are presented as counts and percentage. CGD: chronic granulomatous disease; CVC: central venous catheter; IDU: intravenous drug users.

	Basicranium N = 116	Splanchnocranium N = 334	Other Cranium Bones N = 178	Extremities (No Foot) N = 382	Foot N = 138	Pelvis and Hip N = 175	Ribs and Sternum N = 244	Shoulder N = 155	Vertebrae N = 435
Age, years	59 [2–88]	46 [1–87]	46 [1–87]	38 [0.2–84]	42 [1–92]	41 [0.2–84]	43 [1–84]	36 [9–80]	43 [1–84]
Gender Male Female	70 (60.4) 46 (39.6)	228 (68.7) 106 (31.7)	119 (66.8) 59 (33.1)	273 (71.5) 109 (28.5)	89 (64.5) 49 (35.5)	123 (70.3) 52 (29.7)	185 (75.8) 59 (24.2)	85 (73.9) 30 (26.1)	291 (66.9) 144 (33.1)
Aetiological agent *Aspergillus* *Blastomyces* *Candida* *Coccidioides* *Cryptococcus* *Fusarium* *Histoplasma* *Mucormycosis* *Paracoccidioides* *Phaeohyphomycosis* *Scedosporium* Other	51 (43.9) 6 (5.2) 12 (10.3) - 2 (1.7) 6 (5.2) - 36 (31) - 1 (0.9) 5 (4.3) 3 (2.6)	96 (28.7) 1 (0.3) 21 (6.3) 4 (1.2) 4 (1.2) 7 (2.1) 1 (0.3) 146 (43.7) - - 10 (3) 6 (1.8)	91 (51.1) 13 (7.3) 10 (5.6) 5 (2.8) 11 (6.2) 6 (3.4) 1 (0.6) 22 (12.3) - - 5 (2.8) 2 (1.1)	93 (24.3) 32 (8.4) 36 (9.4) 44 (11.5) 32 (8.4) 13 (2.4) 11 (2.9) 31 (8.1) 30 (7.8) 3 (0.8) 17 (4.4) 24 (6.3)	29 (21) 3 (2.2) 7 (5.1) 24 (17.4) 4 (2.9) 12 (8.7) 1 (0.7) 8 (5.8) 19 (13.8) 1 (9.7) 11 (8) 4 (2.9)	61 (34.9) - 25 (14.3) 31 (17.7) 6 (3.4) - 3 (1.7) 2 (1.1) 20 (11.4) - 1 (0.6) 10 (5.7)	105 (43) 14 (5.7) 25 (10.2) 33 (13.5) 19 (7.8) - 3 (1.2) 1 (0.4) 19 (7.8) - 2 (0.8) 3 (1.2)	48 (41.7) - 5 (4.3) 6 (5.2) 7 (6.1) - - - 28 (24.3) 1 (0.9) 2 (1.7) 1 (0.9)	149 (34.2) 25 (5.7) 134 (30.8) 37 (8.5) 23 (5.3) 7 (1.6) 1 (0.2) 9 (2.1) 19 (4.4) - 10 (2.3) 7 (1.6)
Risk factors Bedsores CGD COVID-19 CVC Diabetes mellitus Disseminated fungal infection Fungemia HIV Hospitalised < 6 months IDU Oncohaematological disease Other immunodepression Parental nutrition Prosthesis Transplanted Surgery/local lesion	3 (2.6) - 3 (2.6) 18 (15.5) 55 (47.5) 28 (24.1) 9 (7.8) - 11 (9.5) - 39 (33.6) 31 (26.7) 4 (3.4) 2 (1.7) 17 (14.6) 24 (20.7)	3 (0.9) 5 (1.5) 55 (16.5) 17 (5.1) 150 (44.9) 50 (15) 19 (5.7) 8 (2.4) 20 (6) 3 (0.9) 57 (17.1) 59 (17.7) 4 (1.2) 2(0.6) 32 (9.6) 64 (19.2)	3 (1.7) - 4 (2.2) 17 (9.5) 40 (22.5) 58 (32.6) 19 (10.7) 7 (3.9) 12 (6.7) 3 (1.7) 47 (26.4) 40 (22.5) 4 (2.4) 3 (1.7) 21 (11.8) 27 (15.2)	5 (1.3) 7 (1.8) - 25 (6.5) 55 (14.4) 102 (26.7) 34 (8.9) 7 (1.8) 45 (11.8) 2 (0.5) 58 (15.2) 84 (22) 10(2.6) 18 (4.7) 43 (11.2) 106 (27.7)	3 (2.2) 1 (0.7) - 18 (13) 24 (17.4) 54 (39.1) 12 (8.7) 1 (0.7) 10 (7.2) 1 (0.7) 34 (24.6) 20(14.5) 4 (2.9) 2 (1.4) 21 (15.2) 16 (11.6)	1 (0.6) - - 10 (5.7) 8 (4.6) 54 (30.9) 22 (12.6) 1 (0.6) 18 (10.3) 1 (0.6) 17 (9.7) 53 (30.3) 8 (4.6) 8 (4.6) 14 (8) 47 (26.9)	- 5 (2) - 3 (1.2) 24 (9.8) 70 (28.7) 19 (7.8) 1 (0.4) 24(9.8) 8 (3.3) 34 (13.9) 54 (22.1) 1 (0.4) 4 (1.6) 19 (7.8) 62 (25.4)	- - - 1 (0.9) 5 (4.3) 40 (34.8) 1 (0.9) 2 (1.7) 6 (5.2) 2 (1.7) 17 (14.8) 23 (20) - - 15 (13) 23 (20)	3 (0.7) 1 (0.2) - 56 (12.9) 53 (12.2) 116 (26.7) 52 (11.9) 4 (0.9) 52 (11.9) 33 (7.6) 81 (18.6) 98 (22.5) 14 (3.2) 5 (1.1) 44 (10.1) 88 (20.2)
Diagnosis Empirical Isolate from bone Histological Other microbiological m. Radiological	1 (0.9) 96 (82.8) 83 (71.5) 2 (1.7) 83 (71.5)	10 (3) 221 (66.2) 243 (62.7) 7 (2.1) 187 (56)	2 (1.1) 135 (75.8) 123 (69.1) 11 (6.2) 111 (62.3)	3 (0.8) 320 (83.8) 266 (69.6) 67 (17.5) 267 (69.9)	2 (1.4) 102 (73.9) 98 (71) 27 (19.6) 84 (60.9)	- 134 (76.6) 119 (68) 33 (18.9) 124 (70.9)	5 (2) 201 (82.4) 177 (72.4) 49 (20.1) 174 (71.3)	- 93 (80.9) 88 (76.5) 2 (1.7) 83 (72.2)	8 (1.8) 365 (83.9) 317 (72.9) 46 (10.6) 321 (73.8)
Time to diagnosis 0–1 month 1–6 months >6 months	36 (31) 17 (14.6) 6 (5.2)	57 (17.1) 96 (28.7) 11 (3.3)	35 (19.7) 53 (29.8) 15 (8.4)	87 (22.8) 101 (26.4) 35 (9.7)	15 (10.9) 40 (29) 4 (2.9)	25 (14.3) 43 (24.6) 10 (5.7)	43 (17.6) 60 (24.6) 15 (6.1)	13 (11.3) 36 (31.3) 3 (2.6)	70 (16.1) 156 (35.9) 30 (6.9)
Outcome Death Recovery Recurrence/chronicity	24 (20.7) 79 (68.1) 10(8.6)	34 (10.2) 225 (67.4) 30 (9)	31 (17.4) 107 (60.1) 23 (12.9)	46 (12) 262 (68.6) 46 (12)	13 (9.4) 78 (56.5) 8 (5.8)	18 (10.3) 102 (58.3) 32 (18.3)	27 (11.1) 156 (63.9) 39 (16)	13 (11.3) 56 (48.7) 20 (17.4)	63 (14.5) 279 (64.1) 59 (13.6)
Treatment Duration, months Antifungal 1 Antifungal ≥ 2 Empirical Surgery	3.7 [0–18], 1 * 80 (69) 53 (45.7) 3 (2.6) 89 (76.7)	4 [0–24], 4 * 207 (62) 166 (49.7) 14 (4.2) 231 (69.2)	4 [0–17], 1 * 124 (69.6) 77 (43.2) 4 (2.2) 108 (60.7)	6 [0–36] 8 * 299 (78.3) 203 (53.1) 12 (3.1) 219 (57.3)	5 [0.2–48] 4 * 69 (50) 47 (34.1) 2 (1.4) 75 (54.3)	3 [0.5–24] 5 * 133 (76) 103 (58.8) 5 (2.9) 79 (45.1)	3 [0–36] 7 * 201 (82.4) 148 (60.6) 6 (2.5) 125 (51.2)	3 [1–15] 1 * 74 (64.3) 50 (43.5) 3 (1.7) 55 (47.8)	4 [0–24] 6 * 345 (70.3) 252 (57.9) 6 (1.4) 240 (55.2)

* Patients who required chronic treatment.

**Table 3 microorganisms-11-01828-t003:** Results of logistic regressions showing association of outcome (death, recovery, chronicity/recurrence) with variables of interest.

Outcome: Death (n = 53)			
Variable	OR	95%CI	*p* Value
Surgical treatment	0.687	[0.376–1.290]	0.23
Treatment duration, months	0.54	[0.411–0.675]	<**0.001**
Time to diagnosis 0–1 month 1–6 months >6 months	0.712 1.541 0.930	[0.382–1.287] [0.863 2.759] [0.391 1.966]	0.27 0.143 0.859
Immunocompromised	1.507	[0.837–2.798]	0.18
Site of infection Basicranium Extremities (except foot) Foot Other cranium bones Pelvis and hip Ribs and sternum Shoulder Splanchnocranium Vertebrae	1.282 0.598 - 1.968 0.412 1.127 0.493 0.794 1.762	[0.423–3.185] [0.293–1.140] - [0.884–4.054] [0.066–1.405] [0.472–2.396] [0.027–2.492] [0.294–1.808] [0.957–3.176]	0.622 0.135 - 0.078 0.232 0.768 0.497 0.613 0.063
Aetiological agents *Aspergillus* *Blastomyces* *Candida* *Coccidioides* *Cryptococcus* *Fusarium* *Histoplasma* *Mucormycosis* *Paracoccidioides* *Phaeohyphomycosis* *Scedosporium* Other	2.69 - 0.682 0.390 0.597 3.135 0.611 1.231 - - 0.792 1.269	[1.498–4.829] - [0.289–1.427] [0.021–1.934] [0.222–1.348] [0.670–11.236] [0.033–3.163] [0.450–2.865] - - [0.185–2.329] [0.291–3.893]	**<0.001** - 0.343 0.364 0.254 0.099 0.638 0.653 - - 0.709 0.709
**Outcome: Recovery (n = 399)**			
**Variable**	**OR**	**95%CI**	***p*** **Value**
Surgical treatment	1.541	[0.913–2.561]	0.98
Treatment duration, months	1.144	[1.060–1.252]	**0.001**
Time to diagnosis 0–1 month 1–6 months >6 months	1.075 1.098 0.704	[0.659–1.771] [0.673–1.808] [0.387–1.335]	0.772 0.619 0.693
Immunocompromised	0.835	[0.507–1.358]	0.472
Site of infection Basicranium Extremities (except foot) Foot Other cranium bones Pelvis and hip Ribs and sternum Shoulder Splanchnocranium Vertebrae	0.847 1.373 7.358 0.684 1.112 0.706 0.933 1.143 0.514	[0.377–2.165] [0.811–2.402] [1.551–131.77] [0.352–1.418] [0.481–3.031] [0.378–1.391] [0.296–4.118] [0.578–2.476] [0.312–0.857]	0.707 0.25 0.051 0.282 0.818 0.292 0.915 0.716 **0.009**
Aetiological agents *Aspergillus* *Blastomyces* *Candida* *Coccidioides* *Cryptococcus* *Fusarium* *Histoplasma* *Mucormycosis* *Paracoccidioides* *Phaeohyphomycosis* *Scedosporium* Other	0.646 3.732 1.003 0.845 1.737 0.458 2.66 0.975 - - 0.895 0.513	[0.389–1.087] [0.754–67.632] [0.561–1.879] [0.303–3] [0.869–3.873] [0.124–2.162] [0.518–48.68] [0.458–2.326] - - [0.379–2.465] [0.203–1.471]	0.094 0.203 0.992 0.768 0.143 0.266 0.349 0.951 - - 0.813 0.179
**Outcome: Recurrence/Chronicity (n = 62)**			
**Variable**	**OR**	**95%CI**	***p*** **Value**
Surgical treatment	1.266	[0.696–2.422]	0.455
Treatment duration, months	1.088	[1.039–1.142]	**<0.001**
Time to diagnosis 0–1 month 1–6 months >6 months	0.896 0.478 3.048	[0.512–1.545] [0.257–0.850] [1.643–5.534]	0.698 **0.014** <**0.001**
Immunocompromised	0.661	[0.385–1.129]	0.129
Site of infection Basicranium Extremities (except foot) Foot Other cranium bones Pelvis and hip Ribs and sternum Shoulder Splanchnocranium Vertebrae	0.363 1.024 0.827 0.975 2.184 1.643 2.953 1.091 1.461	[0.057–1.233] [0.568–1.793] [0.240–2.182] [0.387–2.138] [0.933–4.692] [0.791–3.199] [0.912–8.286] [0.481–2.239] [0.814–2.557]	0.171 0.934 0.73 0.954 0.055 0.16 **0.049** 0.821 0.192
Aetiological agents *Aspergillus* *Blastomyces* *Candida* *Coccidioides* *Cryptococcus* *Fusarium* *Histoplasma* *Mucormycosis* *Paracoccidioides* *Phaeohyphomycosis* *Scedosporium* Other	1.001 0.782 1.206 1.618 0.487 0.665 3.966 0.802 - 1.688 1.215 1.069	[0.537–1.789] [0.122–2.821] [0.614–2.242] [0.453–4.555] [0.182–1.089] [0.035–3.566] [1.184–11.907] [0.268–1.943] - [0.085–11.647] [0.4–3.029] [0.245–3.263]	0.998 0.747 0.567 0.401 0.108 0.7 **0.016** 0.656 - 0.642 0.7 0.916

## Data Availability

No new data were created in this study. Template data collection forms, data extracted from included studies, data used for all analyses and analytic code are available upon reasonable request.

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
