# Peer review of "Fungal Osteomyelitis: A Systematic Review of Reported Cases"

_microorganisms, 2023, doi:10.3390/microorganisms11071828_

Round 1
Reviewer 1 Report
This is a significant study in a very important matter: fungal osteomyelitis. Fungal osteomyelitis is considered a rare disease and the published literature is mainly comprised of case reports, case series and narrative reviews.
The purpose of the paper is clear and important; however, some changes should be done in order to make this work proper to be published.
A PDF is attached marked with important issues.
Changes that I suggest concerns:
Pag 1, Line 12: Where it is “fungal” it should be “Fungal”
Page 2, Line 64: Either you call it gray or grey (see Figure 1).
Pag 2, Line 69-70: In my opinion it is not necessary to describe the contribution of each author on each task. Explain how the worksheet was obtained but not with details about senior and junior contribution.
Pag 5, Line 27: In Figure 2 caption it should be Histogram instead of histogram.
Pag 5, Figure 2b: In my opinion this should be changed into an histogram, graph and not a word cloud.
Pag 5, Table 1: The table does not appear in the right format. It is not readable.
Pag 10, Table2: It was not possible to read this table once is not formatted properly.
Pag 14, Line 262 and throughout all the discussion: Please do not use “we” or “our”. The text should be in the third person or “the authors”.
The subject of the article is very relevant and of immense importance. After the proposed changes, the manuscript should be ready for publication.

Author Response
We thank the reviewer for the time and the dedication spent on our manuscript. We answer their concerns as follows:
Pag 1, Line 12: Where it is “fungal” it should be “Fungal”: Modified as requested.
Page 2, Line 64: Either you call it gray or grey (see Figure 1): Corrected throughout the manuscript.
Pag 2, Line 69-70: In my opinion it is not necessary to describe the contribution of each author on each task. Explain how the worksheet was obtained but not with details about senior and junior contribution: We simplified the explanation of the process removing the distinction between senior and junior researcher but we left the contributions of each author as we feel that the PRISMA guidelines encourage these details.
Pag 5, Line 27: In Figure 2 caption it should be Histogram instead of histogram: Modified as requested.
Pag 5, Figure 2b: In my opinion this should be changed into an histogram, graph and not a word cloud: The graph is now an histogram. The labels have been change accordingly.
Pag 5, Table 1: The table does not appear in the right format. It is not readable: The formatting has been changed. It should now appear correctly.
Pag 10, Table2: It was not possible to read this table once is not formatted properly: The formatting has been changed. It should now appear correctly.
Pag 14, Line 262 and throughout all the discussion: Please do not use “we” or “our”. The text should be in the third person or “the authors: corrected throughout the discussion as requested.
Reviewer 2 Report
The paper is a systematic review of fungal osteomyelitis, and I have to congratulate the authors for the efforts they have made. I think this manuscript needs some amendments before publication, as specified below
Abstract
Line 20: “The majority of patients had Aspergillus infection...”I think that this statement does not match completely the findings of the review. Looking at table 1, Aspergillus ranks first with 26.5 %, but Candida and Mucor are largely represented, with 20.7 % and 16.8 % of infections. Moreover, as stated in the discussion (lines 294) “Candida infections, that are considered relatively common, are reported less frequently than the others.” Please rephrase and add in the abstract also the important role of Candida and Mucor.
Line 23:... while immunosuppression seems to be less relevant...Probably you are right, but important exceptions such as Cryptococcus need to be acknowledged.
Material & Methods
2.1. Data sources and management. You have to specify dates of search (for example, from inception to march 2023).
2.3. Data extraction.Lines 76-77....diagnosis (colture, histology, empiric), indirect mycological diagnostic methods (e.g. cryptococcal and coccidioides antigen tests), radiological evidence of disease,....No mention to biomarkers of fungal infection such as BDG for Candida and Aspergillus (and Pneumocystis, also a rare agent of fungal osteomyelitis), GM for Aspergillus, and a variety of PCR tests on blood specimens (for Candida, Aspergillus, Mucor and other). Tests for a range of fungal biomarkers that do not require an invasive sample-collection procedure have been incorporated into adult clinical practice, and it would be interesting to know if they were applies also for the diagnosis of fungal osteomyelitis.
Results
Line 126: “We judged 83.3% of studies to be of adequate quality”. You have to be more clear on the criteria used to evaluate the methodological quality of included studies . In supplemental table 3, column g, the studies are presented as 1= ACCEPTABLE QUALITY; 0= INSUFFICIENT QUALITY. This is not enough. In the material and methods you said that “Bias in included studies was assessed using Joanna Briggs Institute’s critical appraisal checklists. (lines 81-86). Please provide further supplementary material with this checklist for individual studies.
Discussion
You are talking about Selection bias (line 292), but he methodological quality of case reports and case series is based also on other domains (for example , ascertainment, causality and reporting-see BMJ 2017, Murad et al, Methodological quality and synthesis of case series and case report). Please add some comments basing on the quality assessment.
Lines 290—2 “Our work suffers from several limitations, the most relevant of which is poor quality of existing literature on fungal osteomyelitis: we found almost exclusively case reports and case series, and nearly 20% of these were of inadequate.quality.” As you state in the M&M “studies with inadequate quality were not discarded in order to collect all available data on this rare condition (line 85). I completely agree; nevertheless, I strongly suggest to perform subgroup analyses (or metaregressions) for the outcomes considered according to the methodological quality of included studies.
Author Response
We thank the reviewer for the time and the dedication spent on our manuscript. We have welcomed their suggestions as we believe they improve our work and we answered their concerns as explained below.
Page1, Line 20: Please rephrase and add in the abstract also the important role of Candida and Mucor: We added the contributions of Candida and Mucor with percentages.
Page1, Line 24: while immunosuppression seems to be less relevant...Probably you are right, but important exceptions such as Cryptococcus need to be acknowledged. Due to word count constraints we removed this sentence, but we better specified the role of the HIV in Cryptococcus infection in the discussion.
Page2, 2.1. Data sources and management: You have to specify dates of search (for example, from inception to march 2023). We provided this information in the inclusion/exclusion criteria. Dates were inception to 31st of December 2022.
Page2, 2.3. Data extraction.Lines 76-77....diagnosis (colture, histology, empiric), indirect mycological diagnostic methods (e.g. cryptococcal and coccidioides antigen tests), radiological evidence of disease,....No mention to biomarkers of fungal infection such as BDG for Candida and Aspergillus (and Pneumocystis, also a rare agent of fungal osteomyelitis), GM for Aspergillus, and a variety of PCR tests on blood specimens (for Candida, Aspergillus, Mucor and other). Tests for a range of fungal biomarkers that do not require an invasive sample-collection procedure have been incorporated into adult clinical practice, and it would be interesting to know if they were applies also for the diagnosis of fungal osteomyelitis: We have specified in this paragraph all the types of other diagnostic methods found in the articles (including BDG and GM). In order to better specify their role and to distinguish between species-specific antigen tests and non species-specific tests we also added two sentences in paragraph 3.2 and a paragraph in the discussion.
Line 126: “We judged 83.3% of studies to be of adequate quality”. You have to be more clear on the criteria used to evaluate the methodological quality of included studies . In supplemental table 3, column g, the studies are presented as 1= ACCEPTABLE QUALITY; 0= INSUFFICIENT QUALITY. This is not enough. In the material and methods you said that “Bias in included studies was assessed using Joanna Briggs Institute’s critical appraisal checklists. (lines 81-86). Please provide further supplementary material with this checklist for individual studies. We specified in paragraph 2.4 that risk of bias was assessed only qualitatively and that any item answered “no” in the checklists indicated insufficient quality. The checklists themselves are available in the reference list (references 7 to 10). As requested, we provided a supplementary table (Supplementary table 4) showing quality assessment in details (that is, which items of the checklist scored sufficient or insufficient. We also added in paragraph 3.1 the main reasons why studies were judged of poor quality.
You are talking about Selection bias (line 292), but he methodological quality of case reports and case series is based also on other domains (for example , ascertainment, causality and reporting-see BMJ 2017, Murad et al, Methodological quality and synthesis of case series and case report). Please add some comments basing on the quality assessment. We expanded the discussion on the bias of the case reports and case series, citing Murad’s work and saying explicitly, citing examples, that we found bias in all the four domains, which the checklist we used are able to detect.
Lines 290—2 “Our work suffers from several limitations, the most relevant of which is poor quality of existing literature on fungal osteomyelitis: we found almost exclusively case reports and case series, and nearly 20% of these were of inadequate.quality.” As you state in the M&M “studies with inadequate quality were not discarded in order to collect all available data on this rare condition (line 85). I completely agree; nevertheless, I strongly suggest to perform subgroup analyses (or metaregressions) for the outcomes considered according to the methodological quality of included studies. As suggested, we performed a subgroup association analysis excluding articles of insufficient quality. A commentary on this has been inserted in the main text in paragraph 3.5, while the analysis itself has been added to the supplementary material (Supplementary Table 6) since the results did not change much, probably due to the small number of excluded cases (54).
Round 2
Reviewer 1 Report
The authors attended all the issues therefore the manuscript is now publishable.
Reviewer 2 Report
all the points raised have been addressed. Accept in the present form